https://doi.org/10.1038/s42003-020-1084-0　　**OPEN**
# Nano-imaging trace elements at organelle levels in *substantia nigra* overexpressing α-synuclein to model Parkinson's disease

Laurence Lemelle [1], Alexandre Simionovici [2,3✉], Philippe Colin[4,5], Graham Knott [6], Sylvain Bohic [7,8], Peter Cloetens [8] & Bernard L. Schneider [4,5✉]

Sub-cellular trace element quantifications of nano-heterogeneities in brain tissues offer unprecedented ways to explore at elemental level the interplay between cellular compartments in neurodegenerative pathologies. We designed a quasi-correlative method for analytical nanoimaging of the *substantia nigra*, based on transmission electron microscopy and synchrotron X-ray fluorescence. It combines ultrastructural identifications of cellular compartments and trace element nanoimaging near detection limits, for increased signal-to-noise ratios. Elemental composition of different organelles is compared to cytoplasmic and nuclear compartments in dopaminergic neurons of rat *substantia nigra*. They exhibit 150–460 ppm of Fe, with P/Zn/Fe-rich nucleoli in a P/S-depleted nuclear matrix and Ca-rich rough endoplasmic reticula. Cytoplasm analysis displays sub-micron Fe/S-rich granules, including lipofuscin. Following AAV-mediated overexpression of α-synuclein protein associated with Parkinson's disease, these granules shift towards higher Fe concentrations. This effect advocates for metal (Fe) dyshomeostasis in discrete cytoplasmic regions, illustrating the use of this method to explore neuronal dysfunction in brain diseases.

[1] LGL-TPE, ENS de Lyon, Université de Lyon, CNRS, 69342 Lyon, France. [2] ISTerre, Université Grenoble Alpes, Université Savoie Mont Blanc, CNRS, IRD, IFSTTAR, CS 40700, 38058 Grenoble, France. [3] Institut Universitaire de France (IUF), Paris, France. [4] Brain Mind Institute, Ecole Polytechnique Fédérale de Lausanne (EPFL), 1015 Lausanne, Switzerland. [5] Bertarelli Platform for Gene Therapy, Ecole Polytechnique Fédérale de Lausanne (EPFL), Geneva, Switzerland. [6] Centre of Interdisciplinary Electron Microscopy, Ecole Polytechnique Fédérale de Lausanne (EPFL), 1005 Lausanne, Switzerland. [7] INSERM UA7, Synchrotron Radiation for Biomedicine, STROBE, Université Grenoble Alpes, 38058 Grenoble, France. [8] ESRF—The European Synchrotron, ID16A Beamline, 38043 Grenoble Cedex 9, France. ✉email: alexandre.simionovici@univ-grenoble-alpes.fr; bernard.schneider@epfl.ch

Several neurodegenerative diseases of the central nervous system, including Parkinson's disease, lead to perturbations in the tissue content and distribution of specific metals. In particular, a lot of attention was paid to the level and distribution of iron, which normally contributes to essential cellular functions, including mitochondrial respiration, via its capability to transfer electrons. In vulnerable populations of neurons however, iron dysregulation can have detrimental effects such as the production of reactive oxygen species via the Fenton reaction. Hence, it is critical for neurons and glial cells to tightly control metal metabolism. In the case of iron, this is in large part mediated through interactions with iron-binding proteins and molecules such as transferrin, ferritin, specific transporters, heme, or neuromelanin, which determine iron distribution in cellular compartments and organelles. Genetic defects affecting iron metabolism cause brain diseases, including neurodegeneration with brain iron accumulation[1]. In more common neurodegenerative diseases such as Parkinson's and Alzheimer's diseases, the pathology is also associated with iron overload[2–4]. As several central mechanisms involved in Parkinson's disease, such as mitochondrial dysfunction, are tightly coupled with metal dyshomeostasis[5], it is important to explore perturbations in metal distribution in neurons and glia, with the aim to identify potential causal mechanisms in neurodegeneration[6].

Parkinson's disease is a complex neurodegenerative motor disorder characterized by the degeneration of nigral dopaminergic neurons and by the accumulation and aggregation of the alpha-synuclein protein (α-syn), leading to the formation of Lewy bodies in vulnerable brain regions. Mitochondrial dysfunction, and in particular complex I deficiency, has been identified as a pathophysiological mechanism underlying Parkinson's disease[7]. Genetic associations have shown that the overabundance of α-syn can cause both familial as well as sporadic forms of the disease[8–10]. The α-syn protein can moderately bind Fe(II) and Fe(III) and displays ferrireductase activity[11,12]. Furthermore, interaction with Fe(III) contributes to α-syn oligomer formation[13]. However, the role of this protein in metal dyshomeostasis remains unclear. Abnormal levels of iron and copper have been reported in cases of Parkinson's disease, mainly observed in the substantia nigra, a region of the ventral midbrain which is selectively vulnerable to the underlying pathology[14–17]. These histopathological manifestations of neurodegenerative processes may reflect metal dyshomeostasis linked to mitochondrial and lysosomal dysfunctions[18], ultimately contributing to cell death via mechanisms such as ferroptosis[19–21]. In brain tissue affected by Parkinson's disease, little is known about the cellular compartments that contribute to metal dyshomeostasis, although higher Fe concentrations have been mainly observed in the soluble fraction extracted from the substantia nigra[22].

Developing direct imaging of metals at the subcellular level, in particular that of organelles in the neurons which are vulnerable to disease, would be a major step toward understanding the pathogenic mechanisms involved[23]. Among the analytical approaches developed to investigate physiological heterogeneities of trace elements in brains[24,25], synchrotron radiation-induced X-ray fluorescence (XRF hereafter) is part of the few imaging approaches[26,27], the only non-destructive multi-elemental method of high sensitivity. XRF has been used to analyze Fe traces in brains affected by Parkinson's disease[28], showing that this element is heterogeneously distributed in brain tissues and accumulates in both the neuronal cells and the extra-neuronal substantia nigra tissue of Parkinson's patients. It reaches levels three times higher than in non-affected subjects[29–31]. XRF also showed that in the human brain, Fe is abundantly present in neuromelanin and increases with age[32].

The highest spatial resolution elemental imaging applied to date to neuronal cultures revealed heterogeneous intracellular Fe distributions[33,34]. Neuronal cells show Fe-enriched perinuclear "puncta" (size < 100 nm), corresponding to granules similar to siderosomes. When neuronal cell culture models are exposed to Fe-supplemented media and induced to overexpress α-syn, the Fe-enriched puncta imaged by XRF appear to be enlarged, which indicates perturbed iron homeostasis[35,36]. Aside from the Fe puncta, XRF maps of neuronal cells typically display blurred regions of elemental contrasts in the nuclear and cytoplasmic compartments. Both low elemental count-rates as well as low contrasts at the minimal detection limits of various elements, prevented from extracting robust areas for the elemental maps.

Although challenging, imaging multiple trace elements at organelle level by XRF[37–39] can be tackled and reach the required sensitivity and spatial resolution at third-generation synchrotrons that develop high brilliance hard X-ray nanoprobes exploiting efficient nanofocusing optics and compact ultra-sensitive multiple-element detector systems. Cryo-preparations of neuron cultures were successfully implemented[34,39], and are considered to be the optimal way to preserve the cells in a close-to-native state. In the case of brain tissues and biopsies, alternative approaches have to be considered when targeting a specific region implying the preservation of a large volume and vitrification at an excessive depth[40,41]. Furthermore, coupling to an ultrastructural microscopy method becomes mandatory to unambiguously identify neuronal organelles. This prompted the development of an analytical method based on a transmission electron microscopy (TEM)-nano-XRF imaging correlation, which must be carried out without heavy metal staining usual in preparations for ultrastructural analysis, to avoid elemental contamination impeding XRF analysis. Preparations must also be compliant with the general principles of trace element quantification[42]. In practice, using extremely brilliant nano-beams (>$10^{11}$ ph/s in a 30 nm spot) forces a trade-off between the exposure time for imaging trace elements and the sample radiation damage. Likewise, increasing the sample thickness up to organelle sizes (500–1000 nm) is a trade-off between enhancing the count rates and preserving subcellular spatial resolution.

Here, we introduce an analytical method to measure trace elements in inferred subcellular structures inside neuronal cell bodies located in the substantia nigra tissue. The method is based on quasi-correlative nano-imaging using XRF applied to a 500-nm-thick section of the rat ventral midbrain, and TEM to reveal ultrastructure in an adjacent 80-nm-thin section (Fig. 1). The first survey step of the method consists of (i) displaying histological structures, both in XRF and TEM images, to identify fiducial marks as distant as possible from the region of interest to align these images, and (ii) measuring the elemental content of the cytoplasmic and nuclear compartments. In the second zoom-in step, the elemental composition of the sub-compartments delimited in the ultrastructure TEM images, including organelles, is investigated. This method was applied to several neurons using sections of substantia nigra, revealing elemental compositions specific to different organelle types compared with the nuclear compartment. Next, taking advantage of the highest signal-to-noise ratio (SNR) reached by finely delineating the cytoplasmic regions, we determined changes in the elemental distribution at the highest resolution (down to 20 nm pixel size) following local overexpression of the α-syn protein in the substantia nigra of wild-type rats. In these animals, the accumulation of the human α-syn protein, which is associated to Parkinson's disease, leads to the selective degeneration of nigral dopaminergic neurons[43]. Although the pathology induced in the rat midbrain does not replicate changes observed in the human substantia nigra, such as the formation of Lewy bodies as well as the age-dependent deposition of neuromelanin, this animal model likely mimics the early pathogenic process downstream of α-syn

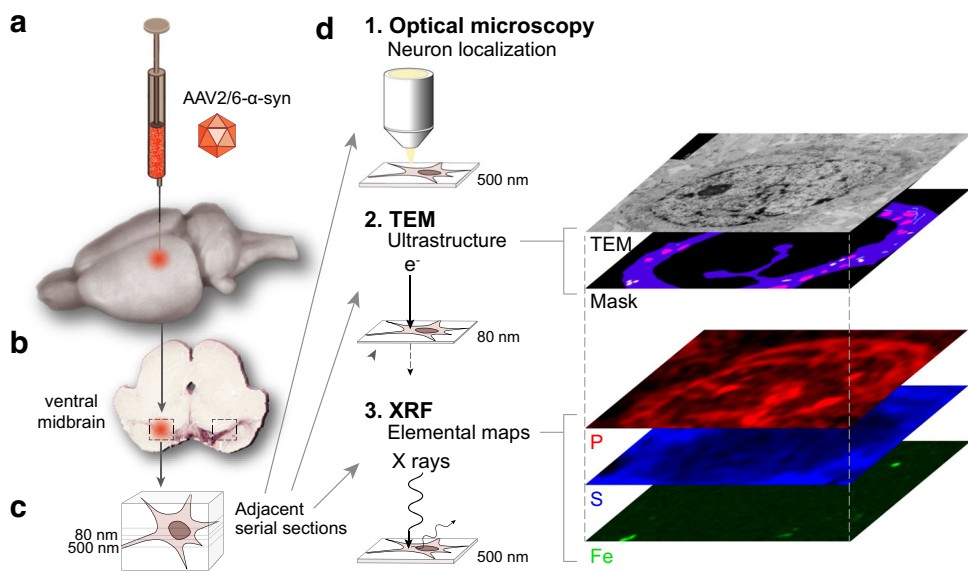

**Fig. 1 Schematics of the method applied for imaging the multi-elemental composition of organelles in nigral neurons overexpressing α-syn. a** AAV2/6 vector encoding human α-syn is unilaterally injected in the rat *substantia nigra*. **b** One month post injection, the ventral midbrain is processed by high-pressure freezing, followed by methanol substitution, resin embedding, and semi-thin sectioning. **c** Consecutive 500 and 80 nm thick sections resistant to radiation damage were obtained using low-temperature embedding of tissues in EPON™ resin applied to preserve the ultrastructure and minimize tissue shrinkage during the resin infiltration. **d** (1) Optical imaging is used to localize neurons in the rat *substantia nigra*. (2) Ultrastructure mosaics are assembled from images recorded on a standard TEM (Tecnai Spirit, FEI Company). (3) XRF images are recorded on the ID16A nano-imaging beamline at ESRF and elemental maps compiled. Masks are defined from TEM images for each type of organelle and matched to the XRF maps of the neuron to calculate the elemental content per organelle type. This quasi-correlative imaging was applied to neuronal cell bodies in either the non-injected or α-syn-overexpressing hemisphere.

accumulation associated with Parkinson's disease. In nigral dopaminergic neurons exposed to the α-syn-induced pathology, nano-imaging resolves the presence of Fe-enriched multi-pixels granules in the cytoplasm. Although better statistics would enhance differences between neurons exposed or not to this pathological condition, the measured Fe and S content of these granules revealed a local shift toward more heterogeneous and higher Fe concentrations in the granules of diseased neurons. Our proposed analytical method provides a tool to assess trace metal dyshomeostasis at organelle levels in an animal model of Parkinson's disease.

## Results

**Nano-XRF visualization of neurons in the *substantia nigra*.** Several histological structures including neuronal cell bodies were revealed in the *substantia nigra* by nano-XRF. An unstained 500-nm-thick section containing a portion of the *substantia nigra pars compacta* was placed on a thin, ultrapure $Si_3N_4$ window, and introduced in the nano-XRF setup of the ID16A beamline at the European Synchrotron Radiation Facility (ESRF)[44] (see Supplementary Fig. 1). Low resolution and fast (low count-rate) XRF maps of the section were first recorded by Scanning X-ray Microscopy at 17 keV (see coarse-scan mode in "Methods"). They showed features, such as blood vessels, as well as folds and scratches on the section that were also seen in optical microscopy, on the same section or in an adjacent section, stained with toluidine blue. This first mapping was used to locate neurons in the XRF maps of *substantia nigra* (Fig. 2a).

This step was followed by a higher resolution scan of neuronal bodies using a 25–50 nm step size (see fine-scan mode in "Methods"). The scanning procedure was optimized to adjust the dose just below our observed sample breakdown threshold (ca. $4 \times 10^7$ photons/s $nm^2$) (Fig. 2b). As expected, the sum spectrum of all pixels measured in the neuronal cell body shows

major contributions from the Si-rich membrane used to support the brain section, and from the Cl-rich EPON™ resin (Fig. 2c). For each spectrum, the Fe and Zn peaks, as well as the P, S, and Ca overlapping ones, were deconvoluted using a fitting procedure with the freely available PyMca software ("Methods")[45].

To obtain elemental maps, the procedure was optimized to define background levels and applied to a set of ~50,000 spectra covering the entire region of a neuronal cell body. This was used to extract the raw count values for each of the elemental $K_\alpha$ and $K_\beta$ lines identified in the spectrum.

Maps of P and S revealed several cellular features, recognizable in the adjacent optical images of the *substantia nigra*. In particular, these maps showed contrasted structures in the neuropil and the glial cells and neuronal cell bodies, in which the nuclear and cytoplasmic compartments could be distinguished (Fig. 2a, b). The Fe map mainly showed enriched structures in the basal lamina of the blood vessels, but also a few hot spots in the neurons that are not observed in the stained section.

**Trace elements in the cytoplasm and nuclei of neurons.** Elemental patterns of large cellular compartments in neuronal cell bodies in the *substantia nigra* were compiled from nano-XRF imaging. Distinct elemental compositions were measured for the cytoplasmic, nuclear, and nucleolar compartments. The elemental profiles measured across the neuronal cell body showed P enrichments and S depletions in the cytoplasmic and nuclear regions, compared with the extracellular region mainly composed of the neuropil (Fig. 3a). Levels of both P and S were lower in the nuclear compartment compared with the cytoplasm, whereas the perinuclear region was characterized by a P-rich contour (Fig. 3a, b). Based on these observations, the P and S raw fluorescence counts were used to delineate the boundaries of the cytoplasmic and nuclear compartments. Of note, elemental

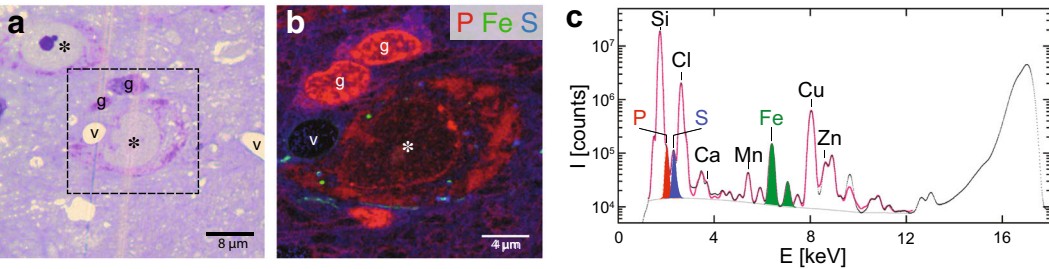

**Fig. 2 Visible and nano-XRF images of *substantia nigra*. a** Optical micrograph of a 500-nm-thick toluidine blue-stained section used to locate large-size neurons in the rat *substantia nigra*. Neurons (*), glial cells (g), and blood vessels (v) can be identified. **b** Nano-XRF image of the neuron (*) and glial cells (g) observed in the dashed frame shown in **a** recorded on a 500-nm-thick adjacent section. This image is extracted from nano-XRF mapping and represents a stack of the P/Fe/S $K_\alpha$ distributions color-coded as red/green/blue, respectively. Acquisition time: 50 ms; pixel size: 50 × 50 nm$^2$; X-ray excitation energy: 17 keV; X-ray fluorescence collection: 12-element Silicon Drift Detector. **c** Sum of the 54510 XRF spectra from the imaged neuron. Note the major XRF contributions of both the 500-nm-thick $Si_3N_4$ membrane holding the section (Si $K_\alpha$ line) and embedding resin (Cl $K_\alpha$ line). The PyMca fit (in pink) also shows the background (dashed grey line) and the $K_\alpha$ peaks of P (red), S (blue), and Fe (green).

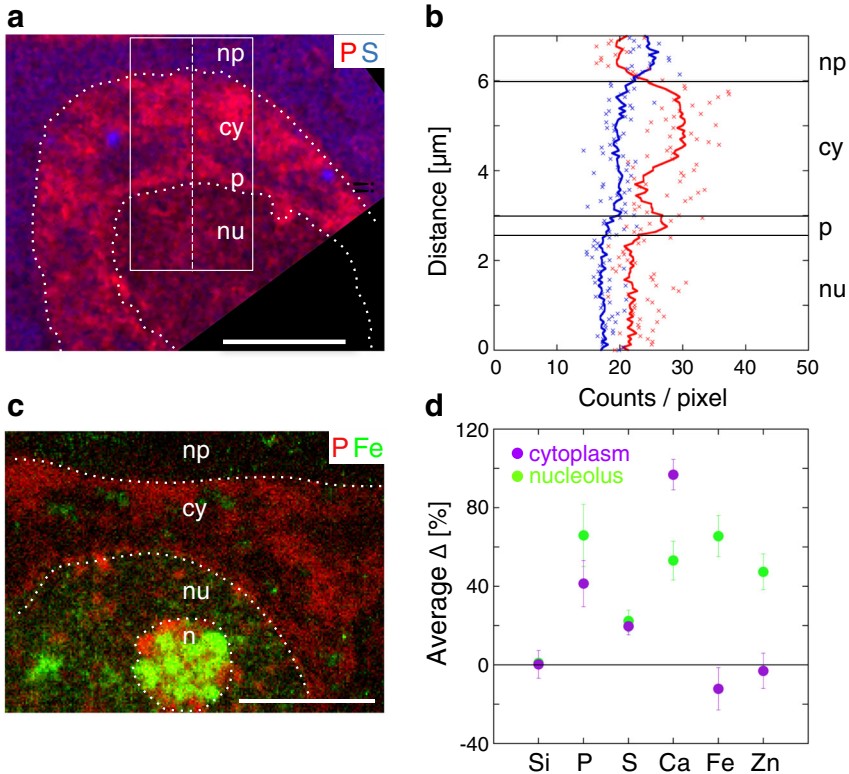

**Fig. 3 Nano-XRF of nuclear and cytoplasmic compartments in neuron bodies. a** Nano-XRF stack of a neuron body (control non-injected condition) showing the P and S maps color-coded as red and blue, respectively. Dotted white lines delimit the nucleus (nu), the nucleolus (n), the P-rich contour of the nucleus (p), the cytoplasm (cy), and the neuropil (np). **b** Raw count XRF profiles (counts/pixel) of P (in red) and S (in blue). Crosses show the P and S raw counts per pixel measured along the 1-pixel-wide dashed line shown in **a**. The P and S counts shown as continuous lines are obtained by averaging values along a 1-pixel-wide horizontal line in the 80-pixel-wide rectangular box displayed in **a**. **c** A nano-XRF stack of a second neuronal body (control condition) showing P and Fe maps color-coded as red and green, respectively. The average Δ for Si, present in the $Si_3N_4$ substrate, is used as an internal control. Scale bars: 4 µm. The XRF acquisitions were carried out at 17 keV with a 6-element detector (Fig. 3a, b) or a 12-element detector (Fig. 3c, d) on two neurons in the rat *substantia nigra* overexpressing human α-syn and two in the control. **d** Average of the relative count rates (Δ) calculated for Si, P, S, Ca, Fe, and Zn in the cytoplasm (76.5 ± 19 µm$^2$ average scanned surface ± SD per neuron, for four neurons) and two nucleoli (5.5 ± 0.8 µm$^2$) with respect to a reference value measured for each element in the nuclear compartment (24 ± 3 µm$^2$). Error bars represent SD values.

imaging of the nucleoli revealed granular structures highly enriched in Fe and P (Fig. 3c).

Next, we compared the elemental composition of these cellular compartments relative to the nuclear area, which was defined as the main part of the nucleus excluding the nucleolus and P-rich perinuclear regions. The P and S contrasts of low SNR per pixel are displayed by an average over all the pixels in a given compartment, $nr_{pixels}$. Elemental contrasts were calculated from the average counts per pixel of the XRF sum spectra of a given compartment. Although the SNR increases, the corresponding minimum detection values decrease so the sensitivity of the elemental contrasts is improved. The relative count rate $\Delta_{i,n}$ of compartment "*i*" with respect to the nuclear compartment "*n*" and the corresponding uncertainty are defined in "Methods."

**Table 1 Elemental areal masses *am*, concentrations *c,* and uncertainties *ε* of Fe, Ca, S, and P, calculated for compartments (nucleus, nucleolus, cytosol) in a neuron body of the *substantia nigra* in the α-syn and control non-injected conditions (CTRL), using an AXO XRF standard. All areal masses, concentrations, and corresponding errors, largely dominated by the AXO standard uncertainties, were compiled by semi-quantitative analysis (see "Methods").**

| | | | Nucleus | | | Nucleolus | | | Cytoplasm | | | FeS-G | |
|---|---|---|---|---|---|---|---|---|---|---|---|---|---|
| | | | *am* ng/cm² | *c* ppm | *ε* (%) | *am* ng/cm² | *c* ppm | *ε* (%) | *am* ng/cm² | *c* ppm | *ε* (%) | *am* ng/cm² | *ε* (%) |
| CTRL | N 1 | Fe | 10 | 200.9 | 14.0 | 14.3 | 286.3 | 13.3 | 7.7 | 154.6 | 13.5 | 11 | 13.6 |
| | | Ca | 1.2 | 24.5 | 16.7 | 1.5 | 29.4 | 13.3 | 2.5 | 50.3 | 13.7 | 2.8 | 14.3 |
| | | S | 59.2 | 1183.8 | 16.9 | 61.8 | 1235.7 | 16.8 | 66.6 | 1332.7 | 16.9 | 98.2 | 16.9 |
| | | P | 79 | 1579.4 | 20.6 | 97.9 | 1958.9 | 20.7 | 112.8 | 2256.3 | 20.7 | 86 | 20.7 |
| | N 2 | Fe | 9.8 | 196.9 | 13.3 | 19.6 | 391.8 | 13.3 | 7.6 | 152.5 | 13.5 | 10.6 | 13.2 |
| | | Ca | 1.2 | 24.6 | 16.7 | 1.9 | 38.3 | 15.8 | 2.5 | 49.2 | 13.7 | 2.7 | 14.8 |
| | | S | 60.7 | 1213.1 | 16.8 | 78.9 | 1577.1 | 16.9 | 72.3 | 1447 | 16.9 | 107.3 | 16.9 |
| | | P | 78 | 1559.5 | 20.6 | 155.1 | 3102.4 | 20.7 | 100.9 | 2018.7 | 20.7 | 78.3 | 20.7 |
| α-syn | N 1 | Fe | 8.9 | 178.6 | 13.5 | 14.3 | 285.6 | 13.3 | 8.5 | 169.1 | 13.5 | 22.9 | 13.5 |
| | | Ca | 1.1 | 22 | 18.2 | 1.9 | 37.6 | 15.8 | 2.2 | 43.2 | 13.7 | 2.6 | 15.4 |
| | | S | 47.9 | 957.3 | 16.9 | 58.2 | 1164 | 16.8 | 60.7 | 1214.7 | 16.9 | 66.1 | 16.9 |
| | | P | 72.8 | 1455.7 | 20.7 | 106.6 | 2131.8 | 20.7 | 97.4 | 1948.9 | 20.7 | 85.5 | 20.7 |
| | N 2 | Fe | 8.7 | 173.4 | 13.8 | 13.6 | 272.5 | 13.2 | 8.8 | 175.4 | 13.5 | 23.3 | 13.3 |
| | | Ca | 1.1 | 21.2 | 9.1 | 1.7 | 34.4 | 11.8 | 1.9 | 38.9 | 13.6 | 2.4 | 12.5 |
| | | S | 52.9 | 1058 | 16.8 | 69 | 1380.7 | 17.0 | 62.7 | 1254.1 | 16.9 | 72.9 | 16.9 |
| | | P | 74.6 | 1492.1 | 20.6 | 141.6 | 2831.9 | 20.7 | 117.4 | 2347.6 | 20.7 | 96.2 | 20.7 |

Given the different sections, cells, and experimental acquisition setups, average contrast values and their standard deviations $\Delta_{i,n}^{ave} \pm SD$ were calculated only for relative uncertainties $\sigma_{\Delta_{i,n}}/\Delta_{i,n} < 30\%$ and at least triplicate measurements.

Doing so, the nucleoli appeared to be markedly enriched in P ($+66 \pm 16\%$), S ($+22.4 \pm 6\%$), Ca ($+53 \pm 10\%$), Fe ($+65 \pm 10\%$), and Zn ($+47 \pm 9\%$) (Fig. 3d). Similarly, the cytoplasm was enriched in P ($+41.5 \pm 12\%$), S ($+19.7 \pm 4\%$) and Ca ($+97 \pm 7\%$) (Fig. 3d). Testing the significance of lower Fe and Zn contrasts, and of different high P, S, and Ca contrasts between the α-syn and the control non-injected conditions (Table 1) would have required statistics that are out of reach of the present study. The corresponding elemental areal masses in ng/cm² (and concentrations in ppm) of the nuclei, nucleoli, cytoplasm in several neuronal cell bodies of the *substantia nigra* are in the ranges of 73–155, 47–107, 1–2.5, and 7–23 ng/cm² for P, S, Ca, and Fe, respectively (Table 1).

**Trace element contrasts of organelles**. The characteristic elemental contrasts in the soma of neurons were further refined using XRF and TEM quasi-correlative microscopy. Neuronal cell bodies were imaged by TEM in sections adjacent to the ones analyzed by XRF. Ultrastructure revealed nuclear and cytoplasmic compartments, as well as specific organelles, such as mitochondria, lipofuscin granules, and the rough endoplasmic reticulum (RER) (Fig. 4a, b, d). Based on TEM mosaic images, specific masks were generated for each type of subcellular compartment (nucleus, cytoplasm) and organelles (mitochondria, lipofuscin granules, RER) (Fig. 4c, e), and applied to the corresponding nano-XRF maps (with an uncertainty of 1 pixel) (Fig. 4f and "Methods"). For the alignment of TEM images and nano-XRF maps, particular angular features observed in the boundaries of cells, blood vessels, cracks in the resin, were used as fiducial marks and chosen as distant as possible from the region of interest. The masks were applied to the nano-XRF maps to determine elemental distributions in each compartment or organelle type, assuming that these regions cover nearly corresponding subcellular structures in the two adjacent sections (Fig. 4f). For each type of organelle, masks were used to calculate an XRF sum spectrum over a number of pixels, a method which improves the SNR. The elemental composition was determined by comparison to reference values measured in the nuclear compartment (Fig. 4g). The values of the elemental contrasts confirmed the elemental enrichments in Ca, P > S for the cytoplasm (Fig. 4g).

Note that the nuclear membrane (defined from the TEM contrast) extends to the nuclear and perinuclear cytoplasmic P-rich region. Nevertheless, the unavoidable uncertainty if one uses only the P/S contrasts to delineate the nucleus/cytoplasm boundary, does not change the average cytoplasmic contrast. Compared with the cytoplasm, lipofuscin granules displayed higher levels of Ca and Fe, whereas the region containing the RER was particularly enriched in P and Ca (Fig. 4g). Mitochondria were enriched in Ca and to a lesser extent in P and S, without noticeable contrast compared with the cytoplasm (Fig. 4g). The Zn chemical contrast was remarkably flat for the different subcellular compartments, except for the nucleoli (Figs. 4g and 3d).

This method reveals the compartment- and organelle-specific distribution of metals within neuronal cells in the brain tissue. However, it is important to consider that the retention of elements imaged in situ depends on the method used for sample preparation, which is based here on methanol for freeze substitution before resin embedding. Indeed, metal retention in presence of solvents depends on local interactions with cellular components which may vary across the different compartments.

**α-syn overexpression alters Fe/S in cytoplasmic granules**. Following overexpression of the α-syn protein associated to Parkinson's disease, we analyzed Fe and S contrasts in nigral dopaminergic neurons to explore possible pathological alterations in iron and sulfur distributions. Nano-XRF imaging revealed the presence of sub-micrometer granules enriched in Fe or S in the neuronal cytoplasm (Fig. 5a, b and Supplementary Fig. 2). These granules were systematically defined by thresholding the Fe and S counts at two standard deviations above the corresponding mean background of counts in the cytoplasm, for both the control non-injected hemisphere, and the AAV-α-syn-injected one. The areal masses compiled from the sum spectrum of all the granules per

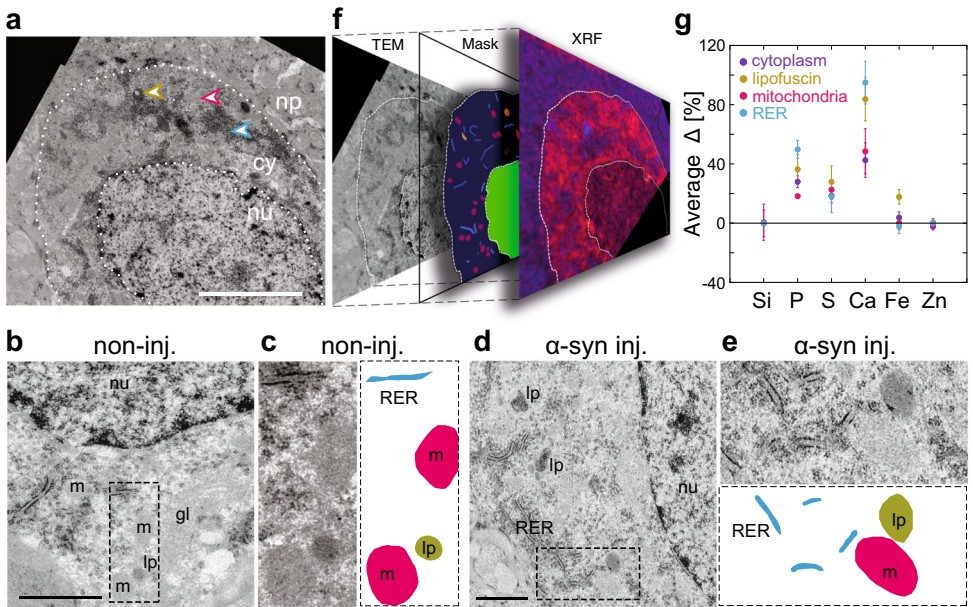

**Fig. 4 TEM and nano-XRF of subcellular compartments in a neuron from the *substantia nigra*. a** Reconstructed mosaic of a single neuron region from TEM images of an ultra-thin serial section stained with uranyl acetate and lead citrate. The nucleus (nu), cytoplasm (cy), and the neuropil (np) compartments also shown in Fig. 3a are annotated. Scale bar: 4 μm. **b** Cropped image across a control neuron (non-injected *substantia nigra*). Scale bar: 2 μm. **c** The region in the dashed frame in **b** is zoomed in and masks corresponding to mitochondria (m, red), RER (blue), and lipofuscin granules (lp, green) are drawn in the adjacent panel. **d** Cropped image across a neuron in the rat *substantia nigra* overexpressing human α-syn (injected with AAV-α-syn). Scale bar: 2 μm. **e** The region in the dashed frame in **d** is zoomed in and masks corresponding to mitochondria (m, red), RER (blue), and lipofuscin granules (lp, green) are drawn in the adjacent panel. **f** Matching the masks of each type of compartment of the neuron to the XRF stack. **g** Average of the relative count rates (Δ) of Si, P, S, Ca, Fe, and Zn calculated from the XRF sum spectrum of the compartments of five neurons (distinct from Fig. 3d): cytoplasm (27 ± 13 μm$^2$), mitochondria (18 ± 2 μm$^2$, 138 structures in the masks), RER (8.6 ± 1.3 μm$^2$, 95 structures), and the lipofuscin granules (2.3 ± 0.3 μm$^2$, 24 structures). The error bars represent SD values. Si is used as an internal control. The XRF acquisition was carried out at 17 keV with a 6-element detector on two neurons in the rat *substantia nigra* injected with AAV-α-syn and three neurons in the non-injected *substantia nigra*.

neuron were found to contain mainly sulfur in the control condition (in the ranges of 98–108 and 10–11 ng/cm$^2$ for S and Fe, respectively, for two neurons shown in Table 1) and to be more enriched in iron in the AAV-α-syn-injected condition (66–72 and 22–23 ng/cm$^2$ for S and Fe, respectively, for two neurons shown in Table 1). The distribution and relative fractions of the Fe and S areal masses of each granule versus its size were further analyzed using the Fiji Analyze Particle plugin[46]. In the AAV-α-syn-injected hemisphere, the granules were found to have on average a larger size (mean and SD area: 0.15 and 0.27 μm$^2$ for 63 granules) than in the control neurons (mean and SD area: 0.08 and 0.1 μm$^2$ for 82 granules) (Fig. 5c). No grain size effects were registered for our 500-nm-thick sections, since comparable low spreads of the data for Fe/S mass ratios were obtained for both homogeneous and inhomogeneous granules of ca. Ø 200 nm average size.

In the non-injected control hemisphere, these granules were found to be rather homogeneous in composition, with a relatively constant Fe/S mass ratio, showing mainly S enrichment (Fig. 5c, d (bottom)), while in the AAV-α-syn-injected hemisphere, a large fraction of the granules were characterized by high Fe content and high Fe/S ratios (Fig. 5c, f (top)). High Fe enrichment was mainly attributed to the granules with sizes ≥ 0.07 μm$^2$ (Fig. 5c) and appeared to be highly inhomogeneous within the granules (see σ/ mean Fig. 5d (middle)).

The XRF maps recorded in the very fine-scan mode ("Methods") showed that in the α-syn condition, the intense Fe signal was often located at the periphery of the largest granules, surrounding a S-rich core (Fig. 5g, bottom). Correlative analysis of XRF with adjacent TEM images showed that in most cases, the Fe/ S-rich granules did not co-localize with any recognizable organelle

structures in the neuronal cytoplasm. However, some of the largest Fe/S-rich granules were found to co-localize with electron-dense organelles, assigned to lipofuscin granules (Fig. 5g, h).

## Discussion

This method quantifies trace elements in identified compartments of neurons of brain tissues. Clear-cut contrasts and higher levels of P and S were obtained in the cytoplasmic compartment as compared with the nucleus, unlike what has been reported either in cell cultures[34,35]. In addition, the perinuclear region showed even higher P levels than the cytoplasm. Although the cause of these differences in P and S levels needs to be further explored, it is worth noting that in the present study, thin sections (500 nm) were probed (see Fig. 3d and Table 1). Hence, the enhanced trace contrasts and subsequent trace quantifications are obtained from uniform matter, either of nuclear or cytoplasmic origin, whereas by imaging thicker sections, it is likely that elemental composition reflects a mix of various subcellular structures.

The nucleolar compartment of nigral neurons, even in non-diseased conditions, displayed clear evidence for the abundant presence of P, S, Ca, but also of Zn and Fe. The presence of Fe in the nucleoli has so far been imaged by XRF in plants[47] and mammalian cells[48], and by other techniques in aged or diseased brains[49,50]. Fe levels in nucleoli of dopaminergic neurons were quantified in the present work (see Table 1). As the *substantia nigra* is a brain region of notoriously high Fe content[22], the possibility that the presence of Fe in the nucleoli might be an in vivo property of dopaminergic neurons calls for further investigations. Besides, the weak overlaps between the Fe and P, Ca, Zn distributions within the nucleolus question the role of metals in close contact with nucleic acids.

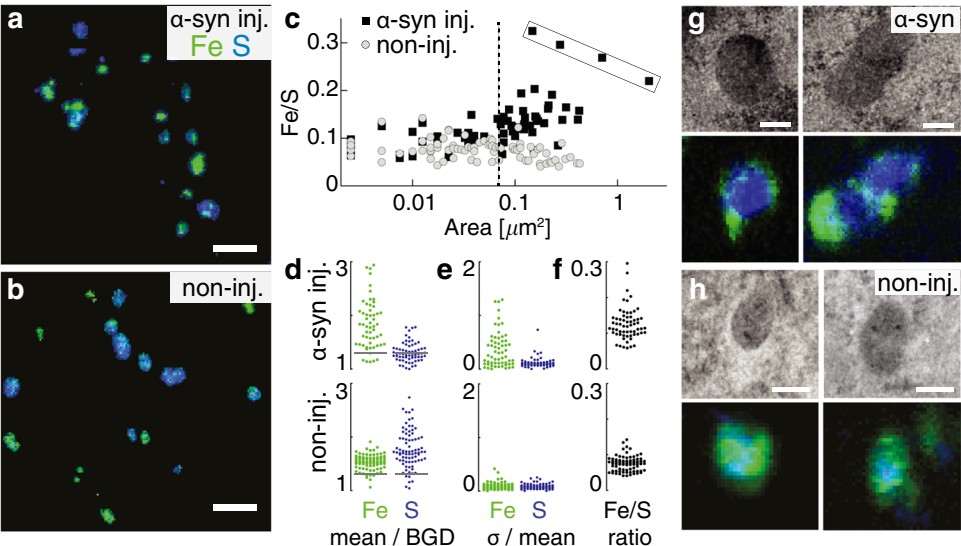

**Fig. 5 Analysis of iron and sulfur-rich granules within the cytoplasm. a, b** Nano-XRF stacks of the Fe and S mass fractions color-coded green/blue in the neuronal cytoplasm in the AAV-α-syn-injected (**a**) and in the control non-injected (**b**) rat *substantia nigra*. Scale bars: 1 μm. **c** Fe/S ratio of the elemental masses versus the granule areas, shown for 63 granules in the cytoplasm of two neurons (AAV-α-syn-injected rat *substantia nigra*, black squares) and for 82 granules in two control neurons (non-injected *substantia nigra*, gray circles). Two size categories are separated by a dotted line (at 0.07 μm²). Note the increase in Fe/S ratio in the AAV-α-syn-injected condition and the presence of a few granules with particularly high Fe/S values (upper rectangle). **d** Dot plots of values of Fe and S content measured within each granule. The graph shows mean ratios of Fe and S over background values (BGD) in the cytoplasm, comparing the control non-injected and the AAV-α-syn-injected rat *substantia nigra*. The grey line displays the threshold used to define Fe and S enrichment. **e** Relative standard deviation (SD) of Fe and S mean values. Note the increased variability in Fe in the AAV-α-syn-injected *substantia nigra*. **f** Fe/S mean ratios for the two conditions showing an increase in the relative Fe/S content in the AAV-α-syn-injected *substantia nigra*. **g, h** Top: higher magnification nano-XRF stacks of individual granules showing S-rich cores surrounded by Fe-rich deposits in the AAV-α-syn-injected (**g**) and in the control non-injected *substantia nigra* (**h**). Bottom: correlation with TEM images shows their partial co-localization with lipofuscin granules. Scale bars: 200 nm. The XRF acquisitions were carried out at 17 keV with a 12-element detector (**a, b**) or a 6-element detector (**g, h**).

The method also displays RERs as particularly enriched in P and Ca, consistent with the role of this organelle in Ca storage, processing of nucleic acids and phospholipid synthesis. Moreover, similar P and Ca enrichments and expected higher Fe levels were observed in lipofuscin granules[51]. High Fe levels in lipofuscins may indicate a role similar to that of the neuromelanin pigment in human dopaminergic neurons, which binds Fe and inactivates toxic Fe cations[52]. By establishing trace metal compositions of the neuronal ultrastructure down to specified organelles, our correlative method provides a tool to further explore metal dyshomeostasis in brain tissue.

The method also showed the existence of Fe-rich granules (diameter < 600 nm) throughout the cytoplasm in the *substantia nigra*, previously observed only in cell cultures[34,36]. Subsets of granules of different Fe/S ratios and sizes, which vary with AAV-α-syn injections, were singled out owing to the better SNR and spatial resolution in uniform and thin preparations. The Fe and S contents of these granules and in particular the shift toward higher Fe concentrations in diseased neurons may indicate that perturbations of iron homeostasis in dopaminergic neurons may play a role downstream of the accumulation of the human α-syn protein following overexpression in the rat midbrain. However, these findings will need to be confirmed in samples of human origin. Remarkably, similar changes in the Fe/S ratio were previously measured at much lower resolution (1 μm) using electron probe microanalysis in both neuronal cells and the neuropil within unfixed midbrain sections of Parkinson's patients[53].

It is therefore important to explore how α-syn accumulation may induce these effects, prompting further research on the role of iron–sulfur clusters in neurodegeneration[54]. Partial co-localization with lipofuscin indicates that Fe-rich granules are likely linked to the deposition of lysosomal material, similar to the co-localization of Fe with neuromelanin observed in human samples. However, most of these Fe/S-rich granules were not as yet matched to any recognizable subcellular structure. Given their small size, the granules observed in the 500-nm-thick XRF sections may not always extend to the adjacent 80-nm-thick TEM sections. Further analysis in thinner XRF sections is therefore warranted, provided these sections can withstand radiation damage. X-ray spectroscopic studies of Fe-speciation and complexation such as those performed in soft X-rays on iron in amyloid plaque cores from Alzheimer's disease subjects would help the identification of these nanostructures and have to be further investigated[55–57]. Similarly, the interpretation of the elemental levels of the mitochondria, close to the cytoplasmic levels, should also benefit from section thinning, if tolerant of higher X-ray doses. Our method quantifies unique multi-elemental nano-heterogeneities at the organelle levels in normal and diseased states, to unfold their contribution to degenerative mechanisms in neurons vulnerable to Parkinson's disease.

In conclusion, we introduce an analytical method to measure trace elemental contents down to organelle levels in dopaminergic neurons within *substantia nigra*. It is based on the quasi-correlative nano-imaging of 500-nm-thick XRF sections and 80-nm-thick adjacent TEM sections of chemically embedded tissues that compensates for the difficulty in targeting specific brain regions and an inability to rapidly vitrify large depths of brain tissue when using only cryo-fixation. To record the highest XRF count rates at the highest spatial resolution, sections of organelle-size thicknesses (ca. 500 nm) were prepared and irradiated up to the damage limits using the highest beam fluences typically withstood by much sturdier thicker sections (>20 μm)[22]. Image processing was carried out by exacting extraction of the low SNR per pixel for each cell (>10⁴ XRF spectra/cell) of the highest fluorescence

signal currently attainable but still close to the limit of detection of our setup. This yielded improved contour maps of the cytoplasm and nuclear compartments, for which elemental average counts per pixel and total loads were compiled. XRF of K-lines using hard X-rays has high trace sensitivity and is the only method to detect few *ppb*[58]. It complements the scope of compositional analyses of organic structures and speciation of minor metal contents linked to them, reachable by soft X-ray correlative microscopy, with a comparable 20–30 nm resolution in 100–200 nm thin tissue sections of brains[26]. Hence, different types of compartments, organelles, or subsets of granules were imaged from which pathological perturbations were discriminated in several neurons.

This method offers considerable opportunities for exploring cellular ultrastructure and function in brain tissues, subject to cautious cytological interpretations. Both the organelles that showed elemental levels very close to the cytoplasm, e.g., the numerous TEM-identified mitochondria in the neurons, and conversely the granules with strong elemental contrasts that do not match any recognizable TEM-ultrastructure, call for further investigations. Testing new sample preparations (varying thickness and fixation) of brain tissues[58] that would better immobilize metals or other bio-essential elements (P, S, K, and Ca)[59], tolerate higher X-rays radiation doses while enhancing TEM contrasts is a prerequisite to increasing the detection contrasts and making them more accurate. Such improvements will be complemented by the next generation of X-ray multi-element detectors coupled with ultra-fast pulse processors allowing the use of a largest fraction of the available intensity of synchrotron beams without saturation. Implementing correlative nano-imaging of XRF with other contrasts is also paramount to constrain biological interpretations[60]. Coupling with X-ray ptychography or phase contrast imaging at the highest spatial resolutions (<20 nm) was shown to be very efficient to reveal subtle elemental differences in single cells, anticipating a finer detection of nano-XRF contrasts in brain tissues[38,40]. The development of new techniques for super-resolution near-field optical imaging (resolution < 100 nm)[61], or other contrasts yielding statistical cytological or biochemical discriminants is a contribution of major importance for diagnosing and exploring the causes of neurodegenerative diseases.

## Methods

**Animal model of Parkinson's disease**. Three 8-weeks old adult female Sprague-Dawley rats weighing ~200 g were housed in standard 12 h light/dark cycles, with ad libitum access to water and food. Anesthetized animals were unilaterally injected in the *substantia nigra* (right hemisphere) with 2 μl of a suspension of the AAV2/6-pgk1-α-syn-WPRE vector encoding wild-type human α-syn protein. The concentration of transducing units (TU) in the vector suspension ($8.35 \times 10^9$ TU/mL) was determined by real-time polymerase chain reaction on total DNA derived from HEK293T cells, 48 h after infection with AAV2/6[62]. The injected vector dose was set at $1.5 \times 10^7$ TU. The vector was injected using a standard stereotaxic procedure according to the following coordinates: −5.2 mm (anteroposterior), −2 mm (mediolateral), −8 mm (dorsoventral, relative to skull surface), −3.3 mm (tooth bar). The injection of this vector was previously shown to induce a local and progressive α-syn pathology in the rat nigrostriatal system[63]. The rats were euthanized 1 month after vector injection. At this time point, there was only mild loss of dopaminergic neurons in the *substantia nigra*, which allowed for the analysis of the perturbations caused by the ongoing accumulation of the human α-syn protein. The *substantia nigra* tissue in the non-injected hemisphere of the same animals was used as control for comparison. All procedures were approved by the Committee for animal experimentation in the Canton de Vaud and performed in accordance with the Swiss legislation and the European Community Council directive (86/609/EEC) regulating care and use of laboratory animals.

**Preparing brain samples for EM and nano-XRF imaging**. Rats were euthanized with an overdose of inhalation anesthetic (isofluorane) and then immediately perfused via the heart with a buffered mix and glutaraldehyde (0.5%) and paraformaldehyde (4%) in 0.1 M phosphate buffer and 100 μm vibratome sections cut in the coronal plane through the region of the *substantia nigra*. The exact region was extracted from the section using a 3-mm diameter biopsy needle and cryo-

protected overnight in 20% sucrose. This was rapidly frozen in a high-pressure freezer (HPM100, Leica Microsystems) and embedded in EPON™ resin at low temperature using methanol as the solvent. EPON™ resin was used for its robustness while cutting at different thicknesses, acrylic resins containing no heavy metal contrast agents (such as uranyl acetate) in the embedding solution, were too fragile to get undamaged cuts of large (1 × 1 mm) semi-thin sections.

Consecutive sections were cut from the final resin-embedded block, with a diamond knife (Diatome, Switzerland) using an ultramicrotome (UC7, Leica Microsystems): a 500 nm followed by a 80-nm-thick slice. The 80-nm-thick slice was mounted on a slot grid for EM and stained with uranyl acetate and lead citrate. The semi-thin section (500 nm) was mounted directly onto the silicon nitride windows (1.5 × 1.5 mm wide 500 nm thick silicon nitride membranes in 5 × 5 mm wide, 200 μm thick silicon frames from Silson Ltd, UK). For nano-XRF, the section was left unstained.

To locate the same tissue region in TEM and nano-XRF, the semi-thin section was first imaged under phase contrast illumination by a light microscope to map the positions of large cell bodies and blood vessels. TEM images of neuronal cell bodies were captured according to their location in this low-resolution map and used to find the same cell in the semi-thin section for nano-XRF imaging. TEM images were taken at a resolution of 1.5 nm per pixel at 80 kV in a (Tecnai Spirit, FEI) TEM and tiled overlapping images were used to capture not only the cell bodies, but also some of the surrounding tissue. On the TEM mosaic images, non-ambiguous compartments were delineated by hand to generate one mask per type of contrast. Translation, rotation, and scaling were applied to align the mask with the corresponding nano-XRF maps using the Fiji TrackEM module.

**Nano-XRF imaging procedures on the ID16A beamline**. XRF measurements were done using the scanning X-ray fluorescence microscopy setup of the ID16A Nano-Imaging beamline of the ESRF (Grenoble)[64]. A pair of multilayer-coated Kirkpatrick-Baez mirrors located 185 m downstream of the undulator source, was used to focus the incident X-rays at the energy of 17 keV down to a 23 nm (horizontal) × 37 nm (vertical) spot size. The specimens were measured under a vacuum of $1 \times 10^{-7}$ mbar at room temperature. They were placed in normal incidence and one or two 6-element silicon drift detectors (SGX Sensortech, UK), according to their availability, were located perpendicular to the beam path on each side of the sample to collect the fluorescence signals. Nanopositioning is performed by a piezo-driven short-range hexapod stage regulated with the metrology of twelve capacitive sensors. All scanning uses "on-the-fly" acquisition with the sample translated at constant speed in the horizontal direction. Low resolution and fast positioning maps were recorded with scan step size fixed at $400 \times 400$ nm$^2$ (dwell time 100 ms, flux ~$1 \times 10^{10}$ photons/s), called coarse-scan mode. Neuron maps were recorded at $50 \times 50$ nm$^2$ fixed scan steps, called fine-scan mode, and details with fine-scan steps fixed at $25 \times 25$ nm, very fine-scan mode (dwell time 50 ms, ~$3 \times 10^{11}$ photons/s in both cases). The summed spectra from the multi-element detectors were fitted using the PyMca software[45].

**Statistics and reproducibility**. The relative count rate $\Delta_{i,n}$ of a compartment "$i$" with respect to the nuclear compartment "$n$" and its uncertainty $\sigma_{\Delta_{i,n}}$ are defined as:

$$\Delta_{i,n} = \frac{(S_i/N_i) - (S_n/N_n)}{(S_n/N_n)} = \frac{S_i}{S_n} \cdot \frac{N_n}{N_i} - 1, \quad (1)$$

$$\sigma_{\Delta_{i,n}} = \left(\Delta_{i,n} + 1\right) \cdot \sqrt{\left(\frac{\sigma_{S_i}}{S_i}\right)^2 + \left(\frac{\sigma_{S_n}}{S_n}\right)^2}, \quad (2)$$

where $S$, $\sigma_S$, and $N$ designate the number of counts, their uncertainties, and the number of pixels, respectively, in the compartments. $S$ and $\sigma_S$ are the output of the PyMca fitting procedure[26]. As a measure of the fluorescence count rate reproducibility, the uncertainty of each pixel, subject to Poisson statistics, is the square root of the number of counts. The number $N$ of pixels in a given compartment thus becomes an important parameter in order to evaluate a $\sigma_{\Delta_{i,n}}$ value with good precision. The organelle/nucleus ratios are typically 100 times less precise than those of cytoplasm/nucleus as the organelle areas are typically ≥ 0.25 μm$^2$ ($10^2$ pixels at 50 nm steps) while the cytoplasm and nucleus have a typical area ≥ 25 μm$^2$ (or $10^4$ pixels at 50 nm steps).

Finally, to qualify the reproducibility of the animal/neuron sampling, SD values were calculated and used throughout.

**Semi-quantitative analyses of elemental areal masses**. Elemental areal masses *am*, and their uncertainties ε, were compiled by semi-quantitative analysis, comparing the ratio of counts in the K$_\alpha$ line of the considered element *X* over that of Si ($S_X/S_{Si}$) to that of the Fe K$_\alpha$ line (in grey in the Table 1) over that of the Si for the AXO Dresden GmbH RF7-200-S2371 thin multilayer standard ($S_{Fe,AXO}/S_{Si,AXO}$). Mounting a 500-nm-thick section of *substantia nigra* on a 500-nm-thick ultrapure Si$_3$N$_4$ window has the advantage of providing an internal Si standard whose analytical drawback is to impair the Si contrast detection. The level of Si in the tissue is indeed negligible with respect to the level of Si measured in the window. The calibration used the average sum spectra per pixel of the

AXO standard, corresponding to 1054 pixels and a total integration time of 105 s. The applied semi-quantitative analytical correction of $am_X$ and $\varepsilon_X$ estimated from the Fe $K_\alpha$ and Si $K_\alpha$ fluorescence lines of samples and from the AXO standard is presented below:

$$am_x = am_{\text{Fe, AXO}} \cdot \frac{f_{\text{Fe, AXO}}}{f_x} \cdot \frac{t_{\text{Si}}}{t_{\text{Si, AXO}}} \cdot \frac{\frac{S_X}{S_{\text{Si}}}}{\frac{S_{\text{Fe, AXO}}}{S_{\text{Si, AXO}}}},$$

$$\epsilon_X = \epsilon_{\text{Fe, AXO}} \cdot \frac{f_{\text{Fe, AXO}}}{f_x} \cdot \frac{t_{\text{Si}}}{t_{\text{Si, AXO}}} \cdot \frac{\frac{S_X}{S_{\text{Si}}}}{\frac{S_{\text{Fe, AXO}}}{S_{\text{Si, AXO}}}}, \quad (3)$$

$$f_X = \frac{\sigma_X \cdot \xi_X}{A_X},$$

where $t$ are the thicknesses ($\text{cm}^{-1}$) of the $Si_3N_4$ sample holder membranes of the $X$ sample and AXO standard and $f$ is the total fundamental parameter correction due to the $\sigma$ fluorescence cross-section ($\text{cm}^2$/g)[65], $\xi$ detector efficiency, and $A$ atomic mass, respectively, of the specific element.

The conversion from $\text{ng/cm}^2$ of the areal masses $am$ to ppm for the $c$ concentration is obtained as follows:

$$c = \frac{am}{t \cdot \rho}, \quad (4)$$

where $t$ is the section thickness, and a value of 1 $\text{g/cm}^3$ was assigned to the density $\rho$ of the chemically fixed brain slices.

**Reporting summary**. Further information on research design is available in the Nature Research Reporting Summary linked to this article.

## Data availability
The datasets generated during and/or analyzed during the current study are available from the corresponding author upon reasonable request.

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

## Acknowledgements

The nano-XRF experiments were supported by the ESRF in the frame of the proposals MD900 and MD970. B.L.S. and Ph.C. were supported by the Swiss National Science Foundation, Grant No 31003A_135696. A.S. and L.L. were supported by the French National Research Agency in the framework of the Investissements d'Avenir program (ANR-15-IDEX-02).

## Author contributions

A.S and B.L.S. designed the research project; G.K., Ph.C., B.L.S. prepared samples and did the TEM work, L.L., A.S., Ph.C., S.B., P.C., and B.L.S. performed the XRF data acquisition; L.L. and A.S. performed the XRF data analyses; B.L.S., L.L., A.S., P.C., and S.B. wrote the paper.

## Competing interests

The authors declare no competing interests.
