## [Peer Review File · Communications Biology]

Reviewers' comments:

Reviewer #1 (Remarks to the Author):

The manuscript describes subcellular trace element quantification in rat brain tissue. There is some excellent and intricate work in this study. The authors use two established imaging methods, TEM and SXRF, with the novelty being the quasi-correlative imaging and measurement of metals in various subcellular compartments, at spatial resolutions that showcase what is technically achievable with hard X-ray nanofocus SXRF beamlines. Brain tissue is evaluated from three rats, where in each animal one hemisphere is treated as the control, and the opposing hemisphere has had injection-induced overexpression of alpha synuclein a period of one month prior to sacrifice. Semi-quantitative measures of the metal concentrations (e.g. Fe, Zn, Ca) are made in the subcellular compartments using SXRF images from 500 nm thick sections informed by TEM images from adjacent 80 nm thick sections.

The study focus is on demonstrating the application of this approach in a relevant system, rather than seeking to unequivocally quantify elemental concentrations in these compartments (as this would not be viable with the sample group size, and the present form of analysis is semi-quantitative). This methodological advance is novel and of interest to the analytical science community. The alpha synuclein aspect of the study provides evidence consistent with iron accumulation in specific compartments in response to over-expression of alpha synuclein, which has scope to be of interest to the medical research community, particularly those considering bioinorganic systems.

The way in which organic materials bind metals can be studied in isolation, and it is presently possible to explore metal ion distributions in cells and tissues. However, technical constraints have been a barrier to detailed descriptions of metal elements associated with subcellular compartments, and may have constrained thinking in the field. The present work may stimulate hypothesis development and testing that has not previously been possible.

The number of examples that could be studied is constrained, this is understandable because of the very limited time that is available for SXRF studies (as it is only available at shared international facilities); I am not aware that this constraint will be overcome for SXRF in the near future, and the reported elemental concentrations need to be treated with caution because of this. However, the report is valuable despite this constraint, it exceeds what can presently be done at these spatial resolutions with alternative techniques, and is likely to underpin future work in the community.

In the present manuscript, there are several places where the order in which information is presented is a limiting factor, or where expansion is required, to improve comprehension and understanding of the work in a wider context. Specific suggestions have been made below.

Recommendations:

1. Title: to consider revising to be explicit that α -synuclein overexpression is the factor making the animal model relevant to Parkinson's disease.
 - This is helpful for those interested in the interactions between metals and peptides;
 - The alpha synuclein overexpression is relevant to a spectrum of neurodegenerative disorders.
2. Abstract line 33: Suggest replacing "would provide" with "offers". This is because technically subcellular trace element measurements can currently be done, as evidenced in the literature (including that cited in the current manuscript). The present study shows how to harness current technical potential to achieve this, rather than inventing a completely new technique.
3. Minor point: Abstract line 42: The grammar is confusing in this sentence, with the comma at

"matrix, Ca-rich" suggesting word(s) are missing.

4. Abstract line 44-46: It is confirmed that overexpression of α -synuclein results in higher concentrations of iron in granules measured in the cytoplasm. However, then it is stated that this points to metal dyshomeostasis and organelle dysfunction. The extrapolation is not explained; perhaps this additional iron is benign accumulation? It would be helpful if the abstract could justify this conclusion.

5. Introduction line 65: Suggest replacing "The direct imaging" with "Developing direct imaging", to implicitly acknowledge the prior work in the field which is cited in subsequent sections of the manuscript.

6. Minor point: Introduction line 67: Grammar: suggest changing to "the pathogenic mechanisms involved."

7. Introduction paragraph 2: This paragraph sets out the contribution of SXRF to identify iron levels in PD brains compared to controls, but in its current form it creates a subjective impression of the field by a) only showing how SXRF has been used, and b) suggesting that studies consistently show higher levels in PD substantia nigra than in controls.

Please can this paragraph be briefly expanded to a) recognise the breadth of work in this area (placing the SXRF in context), and to b) acknowledge that not all studies show an increase in iron in PD. Otherwise, as highlighted by Schrag et al in 2011, this field is particularly vulnerable to publication bias.

8. Introduction lines 75-77: This sentence is about soft X-ray analysis of iron and amyloid in a mouse model of AD, so it doesn't follow naturally from the iron/PD content; perhaps omit this, or additional context is needed? Would need to clarify that this is about neurodegenerative disorders and peptide aggregation in general, and to acknowledge both soft and hard X-ray methods. (Note: currently line 88 specifically mentions hard X-rays and omits soft X-ray microscopy; also see point 23 below.)

9. Introduction line 85: "...elemental compositions could not yet be assigned." – this needs clarification. What was the constraint?

10. Minor point: Introduction line 93: "...and a too significant depth of vitrification" needs correction for grammar. Perhaps "and vitrification at an excessive depth." ?

11. Figures – general points:

- a) Please clarify in captions that these are data from rat substantia nigra (as opposed to simply "substantia nigra", especially as the species is not frequently mentioned in the Results)
- b) Please ensure it is always clear which images and measured values are from injected hemispheres, and which are from un-injected hemispheres.
- c) Please consider labelling image panels with " α -syn injected" instead of " α -syn", to remove the suggestion that the contrast in those panels is a direct measure of alpha synuclein deposition.
- d) Please ensure all data/text within panels is legible, e.g. Figure 4d is difficult to read.

12. Figure 3, line 214: The use of the Si as an internal control, as a primary element in the Si₃N₄ substrate, is mentioned. Is there Si in the tissue, and does this have an implications for the relative count rates (e.g. as reported in Figure 3 part d, also Figure 4)?

13. Minor point: Table 1: is it more useful to have the derived ppm concentrations in the main manuscript and the areal mass values in the SI? (I noted that it is the values in ppm that are reported in the Abstract.)

14. Table 1: footnote for c states "areal masses were compiled by semi-quantitative analysis" – does this note somehow only apply to the nucleus measurements in the first column, or also to the nucleolus and cytoplasm? If to all (which is my understanding), then I think it would be much clearer to simply state this in the table caption rather than have an apparently selective footnote.

15. Results line 296-297: It would be helpful to acknowledge here that retention of elements for imaging will depend on sample preparation – in this case with the use of methanol during embedding - where retention efficiency may be highly compartment-specific depending on how an element is bound.

16. Results line 319: How do the Fe/S ratio findings compare with those in human substantia nigra from controls and PD patients reported by Oakley et al (Neurology 2007)? Does this support the assumption in the present study that the rat with induced alpha synuclein overexpression is relevant to altered iron levels in the human PD brain?

17. Minor point: Figure 5: the blue-green colour scale makes it hard to distinguish Fe and S; perhaps two more distinct colours could be used?

18. Figure 5: is there a possibility that the Fe/S results will be impacted by section geometry as the diameter of the granules approaches section thickness? It would be helpful to include some comment about the constraints with the smaller granules too (currently lines 393/394).

19. Minor point: Discussion line 372: grammar – "displays RERs" better as "displays regions of RER as"

20. Discussion line 379: As this system is modelling Parkinson's disease, and as I assume this animal model does not exhibit detectable neuromelanin pigment, I think this constraint (and the suggested role for lipofuscin) needs to be discussed much earlier than line 379. Ideally in the Introduction, because otherwise the role of neuromelanin in humans, and its absence in the model being used, could limit the perceived clinical relevance of the study.

21. Minor point: Discussion line 399. Grammar: "close to cytoplasm levels, should also benefit from analysis in thinner sections," (also line 422, "cytoplasm levels" instead of "cytoplasm's ones")

22. In the main Methods section, please confirm the age of the three rats. Under the heading Animal Model I believe only the information that they were 'young', and sacrificed one month after alpha synuclein injection, is included.

23. It would be helpful if the authors could add a brief section, perhaps in the Discussion, to place their synchrotron hard X-ray analysis of tissue in context with the opportunities provided by synchrotron soft X-ray analysis. The soft X-ray approach may offer more scope to identify organic structures (perhaps reducing, if not removing, the need for the quasi-correlative work with TEM), but perhaps soft X-ray methods don't offer the same sensitivity or access to the range of chemical elements of interest at the spatial resolutions of interest? A focussed comparison, beyond present comments in the manuscript, would be helpful for those who are not users of these techniques, and it would help confirm the scope for impact of the present study.

Reviewer#2 Editorial Board Member:

Since the general readership of this journal is predominantly biologists, a more detailed information of the disease biology of PD and its relevance to metal (particularly iron) homeostasis at the organelle level would help highlight the importance of the paper.

There should be some comparisons somewhere in the manuscript of this method and other available techniques at least in a semi-quantitative way.

Reviewer#1

1. Title: to consider revising to be explicit that α -synuclein overexpression is the factor making the animal model relevant to Parkinson's disease.
 - This is helpful for those interested in the interactions between metals and peptides;
 - The alpha synuclein overexpression is relevant to a spectrum of neurodegenerative disorders.

The title has been modified and explicitly mentions that our model approach for Parkinson's disease is based on α -synuclein overexpression:

"Nano-imaging of trace elements down to organelle levels in substantia nigra overexpressing α -synuclein to model Parkinson's disease"

2. Abstract line 33: Suggest replacing "would provide" with "offers". This is because technically subcellular trace element measurements can currently be done, as evidenced in the literature (including that cited in the current manuscript). The present study shows how to harness current technical potential to achieve this, rather than inventing a completely new technique.

Done, now p2, line 34

3. Minor point: Abstract line 42: The grammar is confusing in this sentence, with the comma at "matrix, Ca-rich" suggesting word(s) are missing.

Done, now p2 line 43

4. Abstract line 44-46: It is confirmed that overexpression of α -synuclein results in higher concentrations of iron in granules measured in the cytoplasm. However, then it is stated that this points to metal dyshomeostasis and organelle dysfunction. The extrapolation is not explained; perhaps this additional iron is benign accumulation? It would be helpful if the abstract could justify this conclusion.

We thank the Reviewer for this comment and agree that this extrapolation cannot be made in the light of the current knowledge in the field. Therefore, we have changed the wording accordingly (see abstract, now p2, lines 46-48).

5. Introduction line 65: Suggest replacing "The direct imaging" with "Developing direct imaging", to implicitly acknowledge the prior work in the field which is cited in subsequent sections of the manuscript.

Text has been rewritten (see comment 1 of Reviewer #2)

6. Minor point: Introduction line 67: Grammar: suggest changing to "the pathogenic mechanisms involved."

Text has been rewritten (see comment 1 of Reviewer #2)

7. Introduction paragraph 2: This paragraph sets out the contribution of SXRF to identify iron levels in PD brains compared to controls, but in its current form it creates a subjective impression of the field by a) only showing how SXRF has been used, and b) suggesting that studies consistently show higher levels in PD substantia nigra than in controls. Please can this paragraph be briefly expanded to a) recognise the breadth of work in this area (placing the SXRF in context), and to b) acknowledge that not all studies show an increase in iron in PD. Otherwise, as highlighted by Schrag et al in 2011, this field is particularly vulnerable to publication bias.

a) We agree that the paragraph was focused on XRF applied to trace elements in PD. To comply with this request, we have modified the second paragraph of the introduction and

mention recent reviews that put in context the XRF analyses versus other elemental analyses, and XRF imaging versus other imaging techniques.

“Among the analytical approaches developed to investigate physiological heterogeneities of trace elements in brains (Grochowski et al. 2019; Bourassa and Miller, 2012), Synchrotron Radiation induced X-Ray Fluorescence (XRF hereafter) is among the few imaging approaches (Hitchcock et al. 2019, and Collingwood et al. 2017), the only non-destructive multi-elemental method of high sensitivity.”, P4, lines 91-94.

b) We acknowledge that the biases found by Schrag et al. in studies related to Alzheimer’s disease represent a significant concern. However, as we have not found any similar evidence for a bias in the context of Parkinson’s disease, which is the focus of our manuscript, we preferred not to mention it in the manuscript.

8. Introduction lines 75-77: This sentence is about soft X-ray analysis of iron and amyloid in a mouse model of AD, so it doesn’t follow naturally from the iron/PD content; perhaps omit this, or additional context is needed? Would need to clarify that this is about neurodegenerative disorders and peptide aggregation in general, and to acknowledge both soft and hard X-ray methods. (Note: currently line 88 specifically mentions hard X-rays and omits soft X-ray microscopy; also see point 23 below.)

As proposed, this sentence has been removed from p3.

9. Introduction line 85: “...elemental compositions could not yet be assigned.” – this needs clarification. What was the constraint?

We have added the following sentence to provide the main reason why these compositions were not yet assigned:

“Aside from the Fe puncta, XRF maps of neuronal cells typically display blurred regions of elemental contrasts in the nuclear and cytoplasmic compartments. Both low elemental count-rates as well as low contrasts at the Minimal Detection Limits (MDL) of various elements, prevented from extracting robust areas for the elemental maps.” (p4, lines 106-108)

10. Minor point: Introduction line 93: “...and a too significant depth of vitrification” needs correction for grammar. Perhaps “and vitrification at an excessive depth.”?

Done, now p4, line 116

11. Figures – general points:

a) Please clarify in captions that these are data from rat substantia nigra (as opposed to simply “substantia nigra”, especially as the species is not frequently mentioned in the Results)

Done

b) Please ensure it is always clear which images and measured values are from injected hemispheres, and which are from un-injected hemispheres.

Done

c) Please consider labeling image panels with “ α -syn injected” instead of “ α -syn”, to remove the suggestion that the contrast in those panels is a direct measure of alpha synuclein deposition.

Done.

d) Please ensure all data/text within panels is legible, e.g. Figure 4d is difficult to read.

Done

12. Figure 3, line 214: The use of the Si as an internal control, as a primary element in the Si₃N₄ substrate, is mentioned. Is there Si in the tissue, and does this have an implication for the relative count rates (e.g. as reported in Figure 3 part d, also Figure 4)?

The mounting of a 500 nm-thick section of *substantia nigra* on a 500 nm-thick ultrapure Si₃N₄ window has the advantage of providing an internal Si-standard whose analytical drawback is to impair the Si trace contrasts detection. The level of Si in the tissue is indeed negligible with respect to the level of Si measured in the window. (see now p21, lines 576-580 in the “**Semi-quantitative analytical corrections of elemental areal masses**” part of the Methods section).

13. Minor point: Table 1: is it more useful to have the derived ppm concentrations in the main manuscript and the areal mass values in the SI? (I noted that it is the values in ppm that are reported in the Abstract.)

We have introduced in the main body of the manuscript, on page 10, a unique Table displaying both the concentration in ppm, which is the unit used in the medical and biological field, and the areal masses which is the customary parameter in analytical developments.

14. Table 1: footnote for c states “areal masses were compiled by semi-quantitative analysis” – does this note somehow only apply to the nucleus measurements in the first column, or also to the nucleolus and cytoplasm? If to all (which is my understanding), then I think it would be much clearer to simply state this in the table caption rather than have an apparently selective footnote.

This correction was applied in Table 1 and the following sentence was added in the caption.

“All areal masses, concentrations and corresponding errors, largely dominated by the AXO standard uncertainties, were compiled by semi-quantitative analysis (see Methods).” (now p 10, lines 267-268)

15. Results line 296-297: It would be helpful to acknowledge here that retention of elements for imaging will depend on sample preparation – in this case with the use of methanol during embedding - where retention efficiency may be highly compartment-specific depending on how an element is bound.

We thank the Reviewer for this comment and we agree that the use of solvents in sample preparation can lead to bias in the distribution of elements across the different compartments. We have now added the following sentence in the manuscript:

“This method reveals the compartment- and organelle-specific distribution of metals within neuronal cells in the brain tissue. However, it is important to consider that the retention of elements imaged in situ depends on the method used for sample preparation, which is based here on methanol for freeze substitution before resin embedding. Indeed, metal retention in presence of solvents depends on local interactions with cellular components which may vary across the different compartments.” (p13, lines 329-334).

16. Results line 319: How do the Fe/S ratio findings compare with those in human substantia nigra from controls and PD patients reported by Oakley et al (Neurology 2007)? Does this support the assumption in the present study that the rat with induced alpha synuclein overexpression is relevant to altered iron levels in the human PD brain?

We thank the Reviewer for raising this interesting point. We think that the changes in the Fe/S ratio reported in this publication are consistent with our observations at the intracellular level. Therefore, we have now added the following sentence in the manuscript:

“However, these findings will need to be confirmed in samples of human origin. Remarkably, similar changes in the Fe/S ratio were previously measured using electron probe microanalysis in both neuronal cells and the neuropil within unfixed midbrain sections of PD patients [Oakley et al, 2007].” (p16, lines 426-430).

17. Minor point: Figure 5: the blue-green colour scale makes it hard to distinguish Fe and S; perhaps two more distinct colours could be used?

All the images in the manuscript represent the P, Fe, S K_α distributions and were color-coded as RGB, respectively. We believe the same code needs to be used in the manuscript to avoid confusion. However, we provide a different code in the Supplementary Figure 2. We believe this may help readers who cannot easily perceive blue/green contrasts.

18. Figure 5: is there a possibility that the Fe/S results will be impacted by section geometry as the diameter of the granules approaches section thickness? It would be helpful to include some comment about the constraints with the smaller granules too (currently lines 393/394).

The following comment was added:

“No grain size effects were registered for our 500 nm thick sections, as comparable, low spreads of the data for Fe/S mass ratios were obtained for both homogeneous and inhomogeneous granules of ca. Ø 200 nm average size.” (now p13, lines 352-354)

19. Minor point: Discussion line 372: grammar – “displays RERs” better as “displays regions of RER as”

Done, now p16, line 412.

20. Discussion line 379: As this system is modelling Parkinson’s disease, and as I assume this animal model does not exhibit detectable neuromelanin pigment, I think this constraint (and the suggested role for lipofuscin) needs to be discussed much earlier than line 379. Ideally in the Introduction, because otherwise the role of neuromelanin in humans, and its absence in the model being used, could limit the perceived clinical relevance of the study.

Indeed, the rat model used in the present study does not exhibit any neuromelanin pigment. However, the presence of lipofuscin granules in these young animals suggests that the rat nigral dopamine neurons have high lysosomal activity, despite the fact that pigment does not accumulate. The lack of pigment may be due to age and/or differences in dopamine metabolism. Although neuromelanin has been reported to bind Fe (as stated on p4 line 99), it is unclear whether the absence of pigment significantly affects elemental distribution.

To clarify the specifics of the animal modeling approach used in the present study, we have added the following sentences (now p5, lines 142-148):

“Although the pathology induced in the rat midbrain does not replicate changes observed in the human substantia nigra, such as the formation of Lewy bodies as well as the age-dependent deposition of neuromelanin, this animal model likely mimics the early pathogenic process downstream of α-syn accumulation associated with PD. In nigral dopaminergic neurons exposed to the α-syn-induced pathology, nanoimaging resolves the presence of Fe-enriched multi-pixels granules in the cytoplasm.”

21. Minor point: Discussion line 399. Grammar: “close to cytoplasm levels, should also benefit from analysis in thinner sections,” (also line 422, “cytoplasm levels” instead of “cytoplasm’s ones”)

Done, p16, line 448

22. In the main Methods section, please confirm the age of the three rats. Under the heading Animal Model I believe only the information that they were ‘young’, and sacrificed one month after alpha synuclein injection, is included.

Done, p19, line 489

23. It would be helpful if the authors could add a brief section, perhaps in the Discussion, to place their synchrotron hard X-ray analysis of tissue in context with the opportunities provided by synchrotron soft X-ray analysis. The soft X-ray approach may offer more scope to identify organic structures (perhaps reducing, if not removing, the need for the quasi-correlative work with TEM), but perhaps soft X-ray methods don’t offer the same sensitivity or access to the range of chemical elements of interest at the spatial resolutions of interest?

In addition to the brief section added in the introduction (see point 7), we also added a brief comment in the conclusion. It aims to remind that soft X-ray analyses are complementary to those of XRF and will contribute to investigate neurodegenerative diseases.

“XRF of K-lines using hard X-rays has high trace sensitivity and is the only method to reach MDL of few ppb⁵⁸. It complements the scope of compositional analyses of organic structures and speciation of minor metal contents linked to them, reachable by soft X-ray correlative microscopy, with a comparable 20-30 nm resolution in 100-200 nm thin tissue sections of brains » now p17, lines 460-464.

A focused comparison, beyond present comments in the manuscript, would be helpful for those who are not users of these techniques, and it would help confirm the scope for impact of the present study.

P15, line 406, we have introduced a recent reference based on PIXE analyses of the Fe in the nucleoli (Reinert et al. 2019).

P16, line 439-442, we have introduced two recent references based on soft X-rays analyses (Everett et al 2018 and Lermyte et al. 2019) relevant to the analyses of the Fe granules and modified the text accordingly:

“X-ray spectroscopic studies of Fe-speciation and complexation, such as those done in soft X-rays on iron in amyloid plaque cores from Alzheimer’s disease subjects, would help in the identification of these nanostructures and have to be further investigated“

Reviewer#2 (Editorial Board Member):

1. Since the general readership of this journal is predominantly biologists, a more detailed information of the disease biology of PD and its relevance to metal (particularly iron) homeostasis at the organelle level would help highlight the importance of the paper.

We thank the Reviewer for raising this point. We have now added two initial introductory paragraphs to better emphasize the importance of the present study. The added text highlights the fact that changes in the content and distribution of metals is a common feature observed in neurodegenerative diseases. We stress the importance of iron dyshomeostasis to explore the consequences of the dysfunction of mitochondria and other organelles in the context of Parkinson’s disease. In addition, the introduction provides more insight into the interaction between iron and alpha-synuclein, as this pathogenic protein is directly related to the animal model used in the present study. See page 3, lines 56-80.

2. There should be some comparisons somewhere in the manuscript of this method and other available techniques at least in a semi-quantitative way.

According to the shared comments of reviewers #2 and #1 (see points 7, 8 and 23 of rev. #1), we have introduced brief sections in both the Introduction, and the Conclusion parts (p4 lines 91-94 and p17 lines 460-464). These texts provide recent reviews focused on the comparison of the XRF techniques with other techniques. Several new references have also been introduced in the Discussion part and should help the reader to compare semi-quantitatively our results with those obtained with different techniques.

REVIEWERS' COMMENTS:

Reviewer #1 (Remarks to the Author):

My thanks to the authors for their constructive response and for addressing the majority of points in such detail.

I have one main comment on the revision (1) and two minor formatting points (2,3):

(1) The model being used by the authors is hinted at in the title (substantia nigra over-expressing alpha-synuclein), but it is not explicit. I believe it is important to state in the Abstract the fact that this work is in rat brain. Ideally it should be mentioned that the model system is wild-type Sprague-Dawley rat injected with the vector AAV2/6- α -syn encoding human wild-type alpha-synuclein.

The reason I am highlighting this is because the paper is not just reporting an improved methodology, it is also reporting on the properties of the cells. Presently this is being done without explaining what living system these cells are from. Some information can be deduced from Figure 1, but with the Methods at the end of the paper, it is not stated in the main text that these are Sprague-Dawley WT, or explained about the injection with the vector, until line 489 onwards.

(2) Thank you for updating figure labels (Fig 4, Fig 5a,c,d) to make it clear that this refers to an alpha-synuclein injected condition (as opposed to an alpha-synuclein stain); I think Fig 5e also needs the label updated?

(3) Minor point: inconsistent labelling when referring to areal mass. Italic in main text, mix of italic/non-italic in Table 1 caption, and bold non-italic in Table headers.

Reviewer #2 (Remarks to the Author):

All concerns addressed. Thanks.

We thank the Referees for the comments on our manuscript now entitled “Nano-imaging trace elements at organelle levels in *substantia nigra* overexpressing α -synuclein to model Parkinson’s disease”.

The specific comments have been addressed as follows. All the changes are also highlighted in blue in the manuscript.

Reviewer#1

- (1) The model being used by the authors is hinted at in the title (substantia nigra over-expressing alpha-synuclein), but it is not explicit. I believe it is important to state in the Abstract the fact that this work is in rat brain. Ideally it should be mentioned that the model system is wild-type Sprague-Dawley rat injected with the vector AAV2/6- α -syn encoding human wild-type alpha-synuclein.

The reason I am highlighting this is because the paper is not just reporting an improved methodology, it is also reporting on the properties of the cells. Presently this is being done without explaining what living system these cells are from. Some information can be deduced from Figure 1, but with the Methods at the end of the paper, it is not stated in the main text that these are Sprague-Dawley WT, or explained about the injection with the vector, until line 489 onwards.

We thank the Reviewer for this useful comment. We have now edited the Abstract accordingly:

“Elemental composition of different organelle types is compared to cytoplasmic and nuclear compartments in dopaminergic neurons of the rat substantia nigra. They exhibit 150-460 ppm of Fe, with P/Zn/Fe-rich nucleoli embedded in a P/S-depleted nuclear matrix and Ca-rich rough endoplasmic reticula. In-depth cytoplasm analysis displays submicron Fe/S-rich granules, including lipofuscin. Following AAV-mediated overexpression of α -synuclein protein associated with Parkinson’s disease, these granules show a shift towards higher Fe concentrations.”

We hope that it will now be clear to the reader what is the living system analyzed in the present study.

- (2) Thank you for updating figure labels (Fig 4, Fig 5a,c,d) to make it clear that this refers to an alpha-synuclein injected condition (as opposed to an alpha-synuclein stain); I think Fig 5e also needs the label updated?

The figures 4 and 5 as well as the corresponding legends have been edited accordingly.

- (3) Minor point: inconsistent labelling when referring to areal mass. Italic in main text, mix of italic/non-italic in Table 1 caption, and bold non-italic in Table headers.

The text of the manuscript and Table 1 have been modified to address this comment.